# Receptor repertoires of murine follicular T helper cells reveal a high clonal overlap in separate lymph nodes in autoimmunity

Markus Niebuhr[1], Julia Belde[1], Anke Fähnrich[1,2], Arnauld Serge[3], Magali Irla[4], Christoph T Ellebrecht[1,5], Christoph M Hammers[1,6], Katja Bieber[2], Jürgen Westermann[1], Kathrin Kalies[1]*

[1]Institute for Anatomy, University of Lübeck, Lübeck, Germany; [2]Lübeck Institute of Experimental Dermatology, University of Lübeck, Lübeck, Germany; [3]Laboratoire Adhésion et Inflammation, Inserm U1067 CNRS, Aix-Marseille Université, Marseille, France; [4]Centre d'Immunologie de Marseille Luminy (CIML), INSERM U1104, Aix-Marseille Université UM2, Marseille, France; [5]Department of Dermatology, University of Pennsylvania, Philadelphia, United States; [6]Department of Dermatology, University of Lübeck, Lübeck, Germany

*For correspondence:
kathrin.kalies@uni-luebeck.de

Competing interests: The authors declare that no competing interests exist.

**Abstract** Follicular T helper cells (Tfh) are a specialized subset of CD4 effector T cells that are crucial for germinal center (GC) reactions and for selecting B cells to undergo affinity maturation. Despite this central role for humoral immunity, only few data exist about their clonal distribution when multiple lymphoid organs are exposed to the same antigen (Ag) as it is the case in autoimmunity. Here, we used an autoantibody-mediated disease model of the skin and injected one auto-Ag into the two footpads of the same mouse and analyzed the T cell receptor (TCR)β sequences of Tfh located in GCs of both contralateral draining lymph nodes. We found that over 90% of the dominant GC-Tfh clonotypes were shared in both lymph nodes but only transiently. The initially dominant Tfh clonotypes especially declined after establishment of chronic disease while GC reaction and autoimmune disease continued. Our data demonstrates a dynamic behavior of Tfh clonotypes under autoimmune conditions and emphasizes the importance of the time point for distinguishing auto-Ag-specific Tfh clonotypes from potential bystander activated ones.

## Introduction

Germinal centers (GCs) are transient organized microstructures within secondary lymphoid tissues, which support the development of high-affinity antibodies by B cells. Even though GCs are mainly B cell driven, their formation and maintenance depends on follicular T helper cells (Tfh), a specialized subset of effector CD4 T cells. The differentiation into Tfh initiates in T cell zones of lymphoid organs before GC formation starts. After their interaction with B cells at the T-B border, Tfh locate within GC (GC-Tfh) and regulate the survival of proliferating GC-B cells, which compete for antigen (Ag) and for signals from GC-Tfh to undergo further somatic hypermutation and to mature into high-affinity-antibody-producing plasma cells and memory B cells (*Crotty, 2019*; *Qi, 2016*). Thus, GC-Tfh are central players in the regulation of humoral immune responses. Both (i) their differentiation into GC-Tfh and (ii) their survival and clonal expansion within GCs are constantly driven by competition for Ag contacts (*Baumjohann et al., 2013*; *Fazilleau et al., 2009*; *Hwang et al., 2015*; *Knowlden and Sant, 2016*; *Tubo et al., 2013*; *Merkenschlager et al., 2021*).

Here, we asked whether this clonal competition of Tfh for the limited space in GCs involves GC reactions of one or more activated lymphoid organs. This is of special interest for autoimmune conditions when multiple inflamed tissue sites and autoreactive GC reactions exist in distinct lymphoid

organs. Tfh can shuttle between GCs of one lymph node or spleen indicating a local competition within one lymphoid organ (*Merkenschlager et al., 2021*; *Shulman et al., 2013*). The finding that CXCR5+ Tfh circulate in blood (*Brenna et al., 2020*; *He et al., 2013*) opens the possibility for a systemic exchange, in which each Tfh clone could participate in the clonal competition within distinct lymphoid organs in one individual. However, the diversity of individual T cell receptor (TCR) repertoires is extremely high, and it has been shown that individual T cell responses towards Ag are unique and polyclonal (*Textor et al., 2018*; *Thomas et al., 2014*; *Zarnitsyna et al., 2013*). This might be especially valid for auto-Ag because TCR bind with low affinities (*Dolton et al., 2018*; *Rius et al., 2018*). In line, the number of autoreactive T cell clones within one individual is particularly low due to the thymic negative selection process, which could support the involvement of multiple cross-reactive T cell clones specific for the same auto-Ag within one individual as it has been suggested previously (*Ritvo et al., 2018*).

To find out how polyclonal autoreactive Tfh responses are, we studied the distribution of GC-Tfh clones between two activated lymph nodes and challenged the endogenous TCR repertoire by injecting one auto-Ag, directed against a structural protein of murine skin, into the two hind footpads of the same mouse and analyzed the TCRβ sequences of GC-Tfh in both draining popliteal lymph nodes (pln). In contrast to our expectations, we found that the GC-Tfh clonotypes were highly shared between the two pln. This high overlap between both pln was restricted to the dominant GC-Tfh clonotypes and disappeared after establishment of skin pathology even though GC reaction continued. These temporal variations in the distribution of Tfh clones should be considered when utilizing Tfh repertoires to monitor and diagnose autoimmune diseases.

## Results

### GC reactions occur in both draining lymph nodes

To evaluate TCR sequences of GC-Tfh in separate lymph nodes, we used the autoimmune model for the autoantibody-induced skin blistering disease epidermolysis bullosa acquisita (*Hammers et al., 2011*; *Sitaru et al., 2006*; *Figure 1a* and *Figure 1—figure supplement 1*). In this model, autoantibodies against murine type VII collagen (mCOL7), a structural protein of the skin, were induced by injection of the subdomain mCOL7c-GST that was dissolved in phosphate-buffered saline (PBS) and emulsified in the adjuvant Titermax (TM) (herein referred to as Ag1) into the two footpads of one mouse (*Sitaru et al., 2006*). The affinity maturation of B cells in GCs and therewith the development of Ag-specific plasma cells can be observed by the deposition of anti-mCOL7c-specific IgG at the dermal epidermal junction and by the emergence of skin pathology at 2–4 weeks post injection (p.i.) (*Hammers et al., 2011*; *Niebuhr et al., 2020*; *Figure 1—figure supplement 1*).

First, it was important to evaluate that the injection of Ag1 into both footpads would initiate GC reactions in both draining pln. To find out, we stained cryosections immunohistologically for T cells, B cells, and proliferating cells and quantified GC in left and right draining pln by 3D analysis (*Sergé et al., 2015*). The number of GCs varied between 38 and 55 per pln with volumes ranging between 0.005 and 0.01 mm$^3$, which were clearly sufficient to laser-capture 4–6 complete GCs from left and right pln (*Figure 1b* and *Videos 1–4*). Next, we investigated whether the mice in our control group that received Ag-free PBS emulsified in the adjuvant TM only (PBS group) would induce comparable GCs and GC-Tfh as the Ag1 group. As shown in *Figure 1c*, quantification of the GC area within the B cell follicles and T cell numbers within GCs of left and right pln revealed that the GC areas in the PBS group were smaller compared to the Ag1 group (22% for the Ag1 group and 17% for the PBS group), but the numbers of Tfh within the light zone of the GCs did not differ between the Ag1 and PBS groups (*Figure 1d*). To isolate GC-Tfh, we laser-captured an area of 0.04 mm$^3$ GC (4–6 complete GCs, see black dotted line in *Figure 1c*) from cryosections of left and right pln. These laser-captured GCs contained altogether an estimated average of 30,000 GC-Tfh per pln (*Figure 1d*). TCRβ amplification and Illumina Miseq sequencing yielded on average $1.14 \times 10^6$ and $0.08 \times 10^6$ total and 3297 and 2411 unique TCRβ sequences (here referred to as GC-Tfh clonotypes) (*Supplementary file 1*) in the Ag1 and PBS groups, respectively. This data shows that the injections into the two footpads induced GC reactions in both draining pln, which contained comparable numbers of Tfh independent whether Ag is present or not.

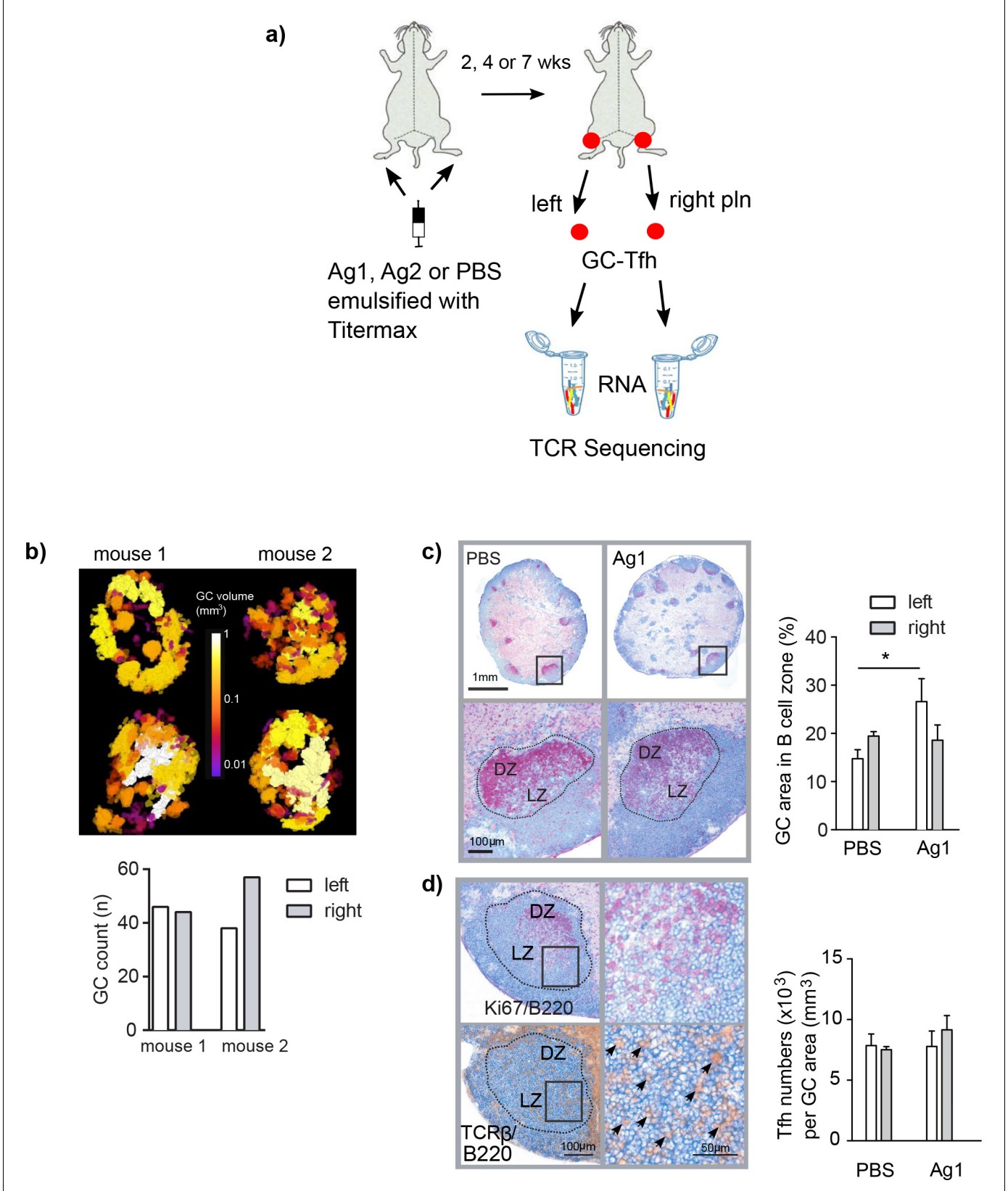

**Figure 1.** Induction of germinal centers (GCs) in left and right popliteal lymph nodes (pln) after immunization with Ag1. (**a**) Experimental setup is shown. Ag1 or Ag2 or phosphate-buffered saline (PBS) emulsified in the adjuvant TM were injected s.c. into both footpads. Both draining pln were isolated 2, *Figure 1 continued on next page*

*Figure 1 continued*

4, or 7 weeks post injection, GCs were laser-captured, RNA was isolated, and TCRβ sequences were separately identified. (b) Serial cryosections of left and right pln were prepared and GCs were stained immunohistochemically with antibodies specific for B cells (B220, blue), proliferating cells (Ki67, red), and T cells (TCRβ, brown). The slides were scanned, and pln were displayed as 3D image (top panel, *Videos 1–4*, GC volumes are shown in the middle). GC volumes of all GCs were calculated and enumerated (bottom panel). (c) Left and right pln were stained immunohistochemically with antibodies specific for B cells (B220, blue) and proliferating cells (Ki67, red, top panel). Dotted lines display the laser-captured GC areas. Magnification shows an area within the GC with the light (LZ) and the dark zones (DZ) (bottom panel). The area of the complete B cell zone and the area of all GCs per cryosection were measured. The GC area was calculated as percentage of the B cell zone area (right). Data are presented as mean ± SD (*p<0.05; n = 3, two pln each, Kruskal–Wallis test). (d) Sequential sections were prepared and stained either for B cells (B220, blue) and proliferating cells (Ki67, red) to identify LZ and DZ or for B cells (B220, blue) and T cells (TCRβ, brown) for detection of Tfh (left panel). Magnification shows an area within the GC (right panel). Arrows indicate individual Tfh. Tfh residing within GCs were counted in left and right pln (right). Data are presented as mean ± SD (no significant differences were found; n = 3, two pln each, Kruskal–Wallis test).

The online version of this article includes the following figure supplement(s) for figure 1:

**Figure supplement 1.** mCOL7c immunization induces autoantibodies, which bind at the dermal epidermal junction and induces skin lesions.

## Identical Tfh clonotypes accumulate in GCs of separate pln

To compare the distribution of GC-Tfh clonotypes between left and right pln, we displayed the identified GC-Tfh clonotypes in regard to their frequency in dot plot diagrams as described (*Gaide et al., 2015*). A high correlation between both pln was found in all mice of the Ag1 group (*Figure 2a*). The dominant GC-Tfh clonotypes especially overlapped almost completely as shown for the 20 most abundant GC-Tfh clonotypes from one representative mouse (*Figure 2b*). This situation was totally different when GC-Tfh clonotypes were correlated to TCRβ sequences of skin-immigrating T effector cells (Teff). In this experiment, a biopsy of an autoantibody-induced inflammatory skin lesions of murine ears was used for TCRβ sequencing (*Supplementary file 2*; *Niebuhr et al., 2020*). We found that the overlap between Tfh and Teff did not exceed the background level (*Figure 2c, d*). To quantify the clonal overlaps between pln and skin lesions for all mice, we used the Morisita–Horn index (MHI), which identifies the number of identical T cell clonotypes between two individual repertoires by taking their relative frequencies into account (*Morisita, 1962*; *Horn, 1966*). Thus, the MHI of GC-Tfh clonotypes between both pln was 3.5 times higher than that between GCs and skin lesions (MHI of 0.46 ± 0.07 versus MHI of 0.13 ± 0.04; *Figure 2d*). This data demonstrates that distinct T cell clones contribute to the responses in GCs and skins, which is in line with data sug-gesting that the binding affinities between TCR and peptide:MHCII complexes determine the direction of T cell differentiation and that only the high-affinity T cell differentiates into Tfh (*Qi, 2016*; *Tubo et al., 2013*; *Cho et al., 2017*).

Next, we asked whether this high overlap between GC-Tfh clonotypes would change when a random set of Tfh residing in the whole left pln would be compared with a random set of Tfh of the whole right pln from the same mouse. Thus, we injected Ag1 into both footpads as described, isolated cells from both pln, sorted CD4+/PD1^high/CXCR5^high Tfh (*Figure 2e*), and determined the individual Tfh TCRβ sequences from 50'000 Tfh cells per pln. 11644 ± 1758 Tfh clonotypes were obtained from which 5781 ± 868 clonotypes were used for analysis (*Supplementary file 3*). Similar to the results obtained from 4–6 laser-captured GCs, a high overlap of Tfh clonotypes between both pln was found (*Figure 2f, g*). Of note, the similarity was even higher in FACS-sorted GC-Tfh clonotypes compared to laser-captured GC-Tfh clonotypes (MHI, 0.46 ± 0.07 versus 0.71 ± 0.11, mean ± SD,

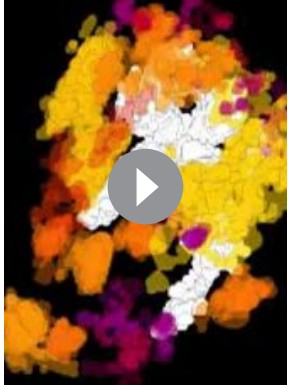

**Video 1.** 3D image of germinal centers (GCs) in the left popliteal lymph nodes (pln) of mouse 1. Ag1/TM was injected into both footpads of SJLH2s mice. A complete collection of serial cryosections of the left pln (4 weeks post injection) was imaged by automatic scanning microscope and processed by For3D. GCs were segmented by filtering, thresholding, and soothing the stack of pln section images.

https://elifesciences.org/articles/70053#video1

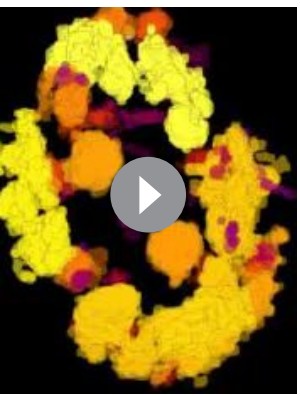

**Video 2.** 3D image of germinal centers (GCs) in the right popliteal lymph nodes (pln) of mouse 1. Ag1/TM was injected into both footpads of SJLH2s mice. A complete collection of serial cryosections of the right pln (4 weeks post injection) was imaged by automatic scanning microscope and processed by For3D. GCs were segmented by filtering, thresholding, and soothing the stack of pln section images.
https://elifesciences.org/articles/70053#video2

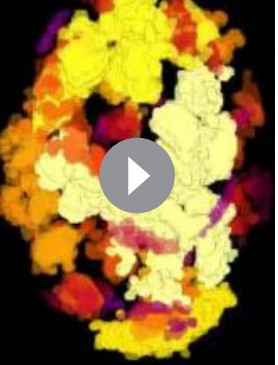

**Video 3.** 3D image of germinal centers (GCs) in the left popliteal lymph nodes (pln) of mouse 2. Ag1/TM was injected into both footpads of SJLH2s mice. A complete collection of serial cryosections of the left pln (4 weeks post injection) was imaged by automatic scanning microscope and processed by For3D. GCs were segmented by filtering, thresholding, and soothing the stack of pln section images.
https://elifesciences.org/articles/70053#video3

*Figure 2f*), indicating that each GC within one pln might harbor the same Tfh clonotypes.

To find out whether this synchronization of GC-Tfh clonotypes is a general principle, independent of Ag or mouse strain, we injected another disease-inducing domain of mCOL7, the vWFA2 subdomain, which is devoid of the GST tag and has a smaller size (190 aa), into the left and right footpad of another mouse strain carrying the identical MHCII haplotype as SJL mice: C57BL/6 H2s mice (herein referred to as Ag2) (*Iwata et al., 2013*). A comparable high similarity of GC-Tfh clonotypes was found (*Supplementary file 4*, *Figure 3a, b*). To test for Ag specificity of the GC-Tfh clonotypes, we compared the V and J gene segment usage between the two Ag groups. We found that injection of the Ag1 in SJL mice induced an accumulation of Tfh expressing TRBV3, whereas injection of Ag2 yielded a significant higher percentage of Tfh bearing the TRBV2 segment in C57BL/6 H2s mice (*Figure 3c*). In addition, an increased percentage of Tfh expressing TRBJ2-5 were found in SJLH2s mice exposed to Ag1. Because Ag1 and Ag2 are presented in the same MHCII haplotype, it is reasonable to conclude that the differences in the TRBV and TRBJ gene segments between both mouse strains are caused by the Ag, which supports the data that GC-Tfh are specific for the immunizing Ag.

## The abundance of the initially dominant GC-Tfh clonotypes declines over time

The maintenance of GC-Tfh requires a constant antigenic stimulation (*Baumjohann et al., 2013*; *Merkenschlager et al., 2021*). Considering that Ag might be consumed over time, we asked whether the highly dominant GC-Tfh clonotypes would remain enriched in both pln. To find out, we analyzed two time points, 2 and 7 weeks after injection of Ag1. The first time point reflects the initial appearance of high-affinity autoantibodies as judged by their binding at the dermal epidermal junction (*Hammers et al., 2011*; *Niebuhr et al., 2020*). At the later time

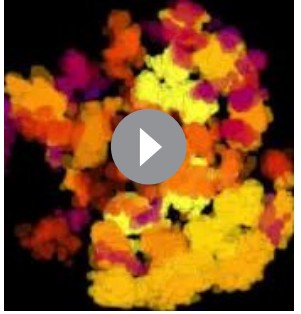

**Video 4.** 3D image of germinal centers (GCs) in the right popliteal lymph nodes (pln) of mouse 2. Ag1/TM was injected into both footpads of SJLH2s mice. A complete collection of serial cryosections of the right pln (4 weeks post injection) was imaged by automatic scanning microscope and processed by For3D. GCs were segmented by filtering, thresholding, and soothing the stack of pln section images.
https://elifesciences.org/articles/70053#video4

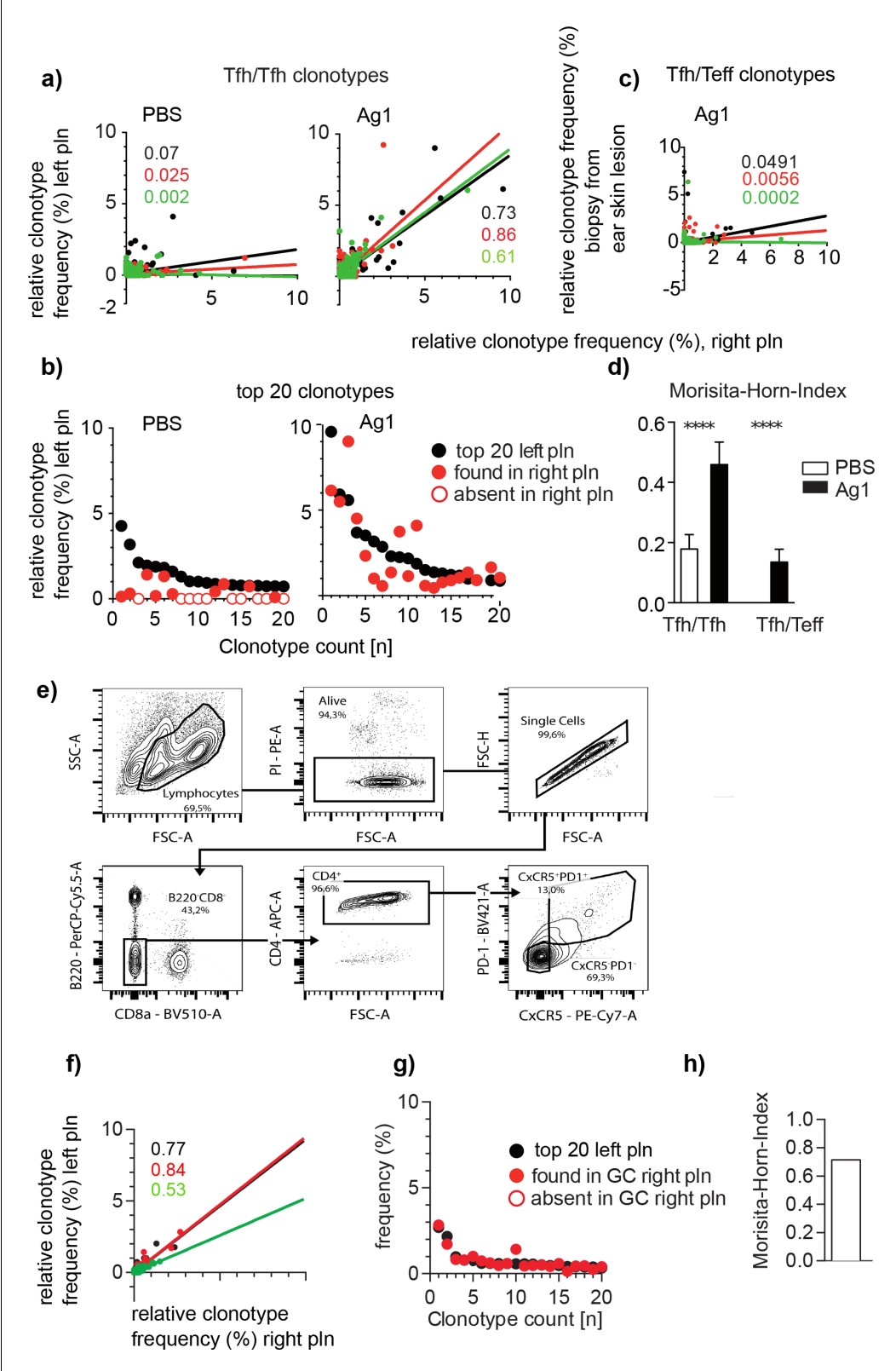

**Figure 2.** Identical Tfh clonotypes accumulate within germinal centers (GCs) of both popliteal lymph nodes (pln) at the time point of skin lesion onset. Ag1 or phosphate-buffered saline (PBS) were injected s.c. into both footpads. GCs from left and right pln were isolated and GC-Tfh clonotypes were analyzed 4 weeks post injection. (a) The frequency of overlapping GC-Tfh clonotypes of the left (y-axis) and right pln (x-axis) is shown as dot plots. Each dot represents one GC-Tfh clonotype. Linear regression lines ranging from 0 to 10 and r (*Qi, 2016*) values are depicted for each mouse (n = 3, mouse

*Figure 2 continued on next page*

*Figure 2 continued*

1: black; mouse 2: red; mouse 3: green). (**b**) The 20 most frequent GC-Tfh clonotypes present in the left pln (black dots) are compared to their presence in the right pln (red dots). One representative comparison out of three is shown. (**c**) The frequency of ear skin-derived TCRβ clonotypes (Teff, y-axis) shared with the GC-Tfh clonotypes of one pln (x-axis) is shown as dot plots. Each dot represents one clonotype. Linear regression lines ranging from 0 to 10 and r (*Qi, 2016*) values are depicted for each mouse (n = 3, mouse 1: black; mouse 2: red; mouse 3: green). (**d**) Data of GC-Tfh clonotypes shared between left and right pln are shown as Morisita–Horn index (MHI; right) and compared to the MHI of similar TCRβ clonotypes between pln and ear skin (mean ± SD; \*\*\*p<0.001, Mann–Whitney U-test, n = 3, two pln each). (**e**) Gating strategy for CD4/PD1/CXCR5 for sorting of Tfh from left and right pln .(**f–h**) as described in (**a, b, d**).

The online version of this article includes the following figure supplement(s) for figure 2:

**Figure supplement 1.** Comparison of MiTCR and MiXCR analysis tools reveals a similar distribution of dominant Tfh clonotypes.

**Figure supplement 2.** Assessment of Tfh clonotypes within left and right popliteal lymph nodes (pln) after injection of glutathione-s-transferase (GST).

point, when disease activity is severe and skin wounds cover more than 8% of the body surface, it is doubtless that high B cell-affinity maturation had occurred. As described above, GCs were laser-captured from left and right pln, the number of Tfh was estimated by counting, and TCRβ sequences were assessed by Miseq illumina sequencing (*Supplementary file 1*). Comparing the frequency of individual Tfh clonotypes from both 2 weeks pln and both 7 weeks pln in dot plot diagrams, a strong positive correlation between both pln was found only after 2 weeks p.i. but not after 7 weeks p.i. (*Figure 4a*). Accordingly, the similarity index MHI decreased from 0.35 ± 0.08 (mean ± SD) at 2 weeks p.i. to 0.24 ± 0.08 at 7 weeks p.i. (mean ± SD) (*Figure 4b*). This data demonstrates that the overlap of the dominant GC-Tfh clonotypes especially diminishes during the period from 2 weeks to 7 weeks p.i., even though the general frequency distribution of all GC-Tfh clonotypes did not change within this period (data not shown).

To find out whether the initially dominant overlapping Tfh clonotypes would completely disappear after chronic disease manifestation, we compared the Tfh repertoire within one mouse at two distinct time points. Therefore, we surgically removed one pln at 2 weeks p.i. The pln from the contralateral side remained in vivo until the end of the experiment (10 weeks p.i.) (*Ellebrecht et al., 2016*). GCs were isolated, TCRβ sequences determined (*Supplementary file 1*), and the distribution of Tfh clonotypes regarding their frequency was displayed in dot plot diagrams as described above.

The results obtained after this surgical removal of one pln for a period of 8 weeks is in line with our data found for distinct mice (*Figure 4a*). Dot plot diagrams of the identified TCRβ sequences from the same mouse showed a low correlation only (*Figure 4c*). However, the analysis of the distribution of the top 20 clonotypes revealed that the majority of the initially dominant 2 weeks GC-Tfh clonotypes was still present in 10 weeks GCs, although at lower frequencies. This situation was different for the dominant GC-Tfh clonotypes of the 10 weeks GCs, which were mostly absent in the surgically removed 2 weeks pln (shown for one representative mouse, *Figure 4d*). Thus, the relative intersection of the top 20 2 weeks GC-Tfh clonotypes and all 10 weeks GC-Tfh clonotypes was significantly higher (55%) than the top 20 10 weeks GC-Tfh clonotypes and all 2 weeks GC-Tfh clonotypes (31%) (*Figure 4e*). This data shows that the dominant GC-Tfh clonotypes, present prior to disease onset, are maintained at lower frequencies during chronic disease manifestation.

## GC-Tfh clonotypes carrying the TRBV3 segment are found in all mice of the Ag1 group

Our finding demonstrates that Tfh clonotypes highly overlap in two distinct pln of the same mouse in response to the auto-Ag. Thus, the question arises, how many of these GC-Tfh clonotypes would overlap with two or all three cage-matched littermates of the same group? To address this question, we determined the number of overlapping Tfh clonotypes within each group by calculating the sharing index as described (*Madi et al., 2014*; *Madi et al., 2017*). The number of all unique Tfh clonotypes per mouse were identified by combining the individual Tfh clonotypes of both left and right pln 4 weeks p.i. Based on this, the number of overlapping Tfh clonotypes in either one, two, or three mice of the same group was quantified. We found that 3.6 more Tfh clonotypes overlapped in the Ag1 group (65) in comparison to the PBS group (18) (*Figure 5a*). These higher numbers of overlapping Tfh clonotypes in the Ag1 group compared to the PBS group were confirmed in Venn diagrams (*Figure 2—figure supplement 2*). Analysis of the TRBV gene segment usage revealed that especially GC-Tfh clonotypes bearing TRBV3 accumulated within GCs (*Figure 5b*). In accordance with the

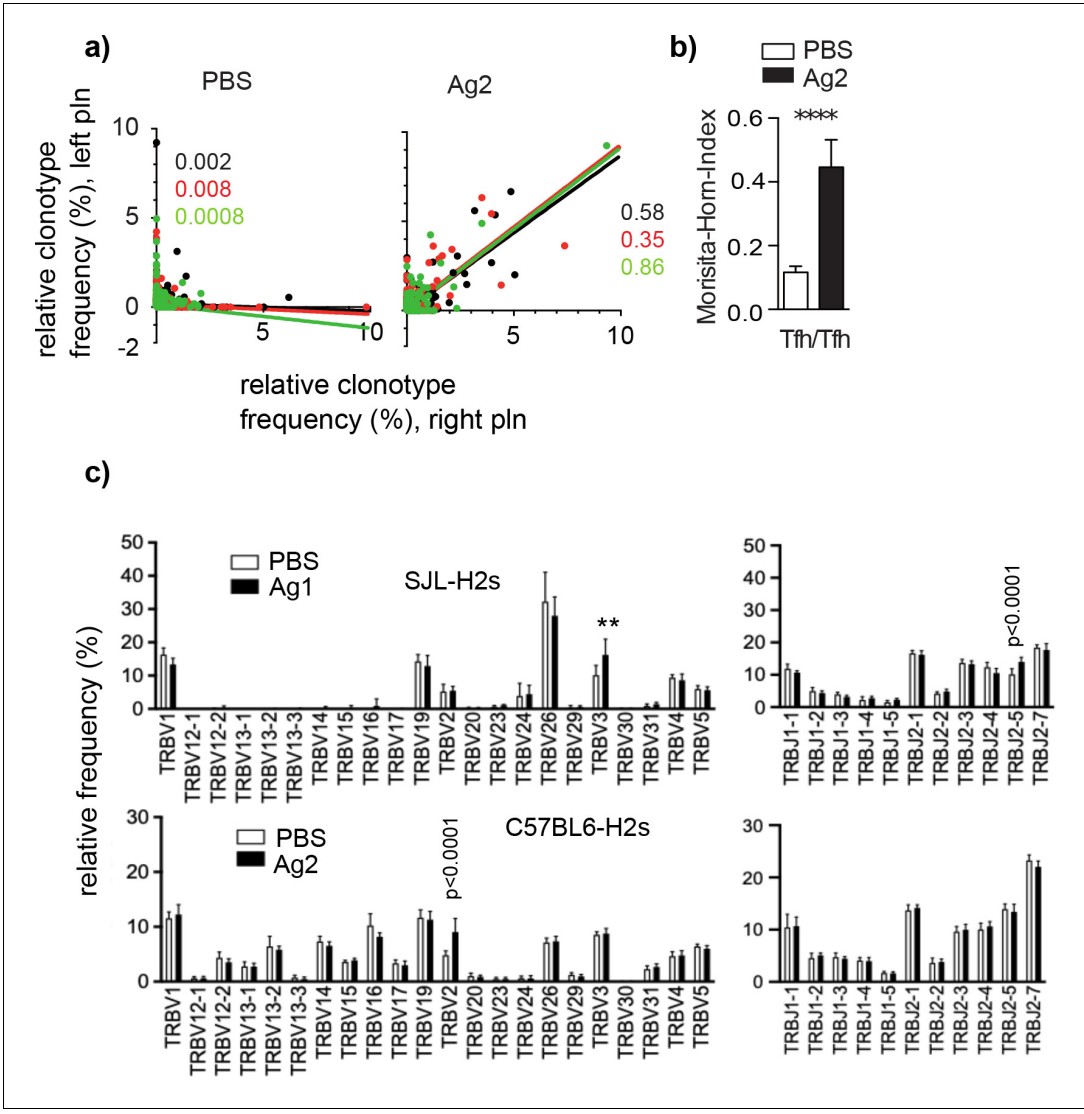

**Figure 3.** GC-Tfh expressing a particular TRBV or TRBJ segment accumulate in an Ag-specific manner. Ag2 or phosphate-buffered saline (PBS) were injected into C57BL/6s mice. Germinal centers (GCs) from left and right popliteal lymph nodes (pln) were isolated and Tfh clonotypes were analyzed 4 weeks post injection. (a) The relative frequency of overlapping GC-Tfh clonotypes of the left (y-axis) and right pln (x-axis) is shown as dot plots. Each dot represents one GC-Tfh clonotype. Linear regression lines ranging from 0 to 10 and r2 values are depicted for each mouse (n = 3, mouse 1: black; mouse 2: red; mouse 3: green). (b) The similarity index (Morisita–Horn index [MHI]) reveals a high overlap in Ag2-exposed pln but not in the PBS group (mean ± SD; ***p<0.001, Mann–Whitney U-test, n = 3, two pln each). (c) Relative frequencies of TRBV (left) and TRBJ (right) expressed by GC-Tfh clonotypes were compared between the Ag and PBS groups after injection of Ag1 (upper panel) or Ag2 (lower panel). Mean ± SD; **p<0.01, p<0.0001, two-way ANOVA with Sidak's correction test, n = 3 (two pln each). Of note, the lack of TRBV segments is caused by deletions in the TCR gene of SJL mice (*Behlke et al., 1986*). The online version of this article includes the following figure supplement(s) for figure 3:

**Figure supplement 1.** Assessment of TRBV frequencies of Tfh clonotypes within germinal centers (GCs) after injection of glutathione-s-transferase (GST).

finding that the number of overlapping Tfh clonotypes decreases between 7 and 10 weeks p.i. (*Figure 4*), the number of overlapping Tfh clonotypes in the Ag1 group dropped to 9 at 7 weeks p.i. (data not shown). Interestingly, four of these nine Tfh clonotypes carried TRBV3. This data supports the notion that TRBV3 bearing Tfh recognize Ag1-derived epitopes presented in MHCII H2s complexes in this skin-blistering autoimmune disease model.

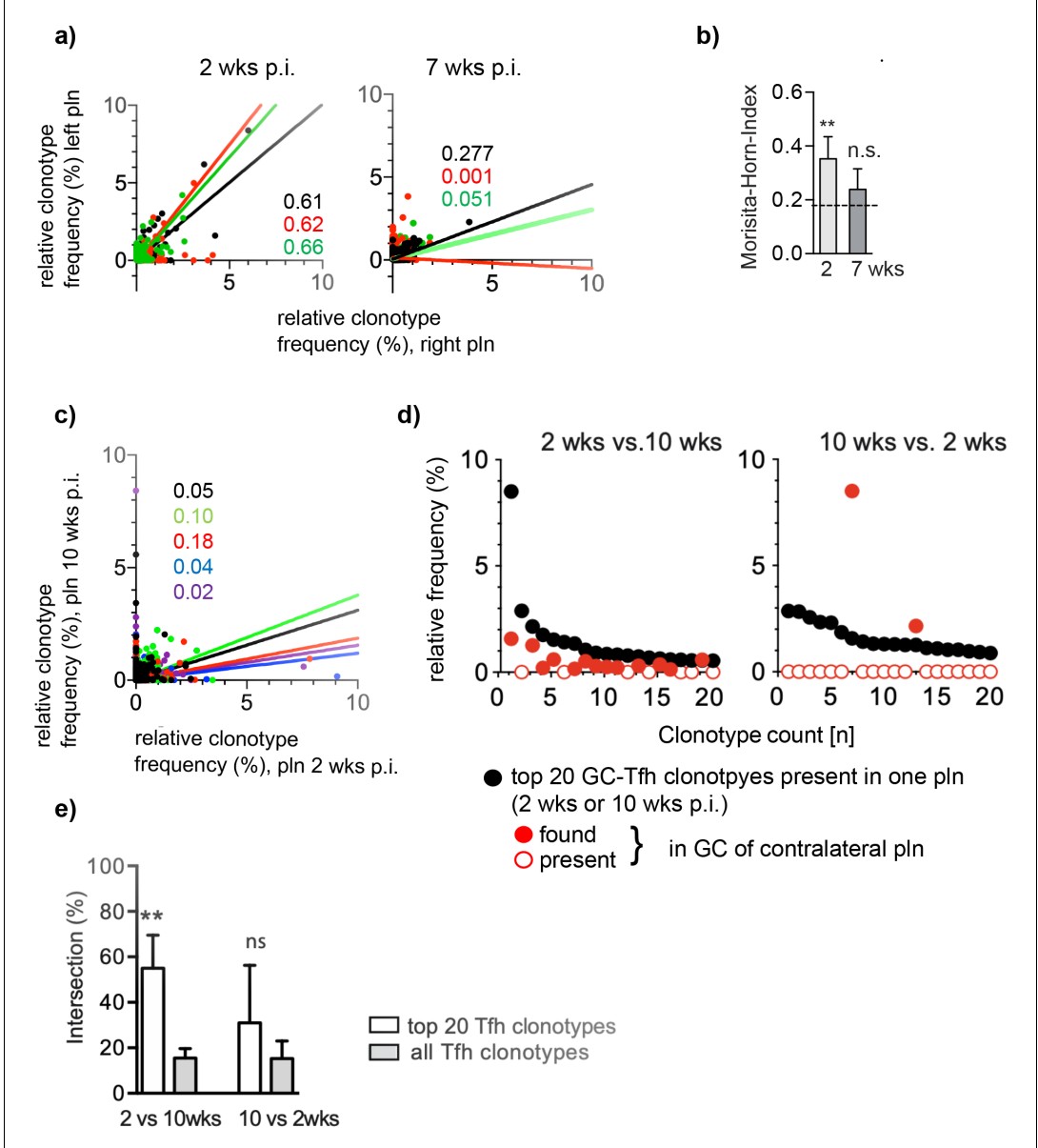

**Figure 4.** The overlap of the dominant Tfh clonotypes in left and right popliteal lymph nodes (pln) disappear over time. Ag1 or phosphate-buffered saline (PBS) were injected s.c. into both footpads. Germinal centers (GCs) from left and right pln were isolated 2 weeks or 7 weeks post infection (p.i.) and GC-Tfh clonotypes were analyzed. (a) The frequency of Tfh clonotypes shared in GCs of the left (y-axis) and right pln (x-axis) at 2 weeks p.i. (left panel) or 7 weeks p.i. (right panel) is shown as dot plots. Each dot represents one GC-Tfh clonotype. Linear regression lines ranging from 0 to 10 and r2 values are depicted for each mouse (n = 3, mouse 1: black; mouse 2: red; mouse 3: green). (b) Data of GC-Tfh clonotypes shared between left and right pln are shown as Morisita–Horn index (MHI) (mean ± SD; **p<0.01, Mann–Whitney U-test, n = 3, two pln each). (c) The left pln (x-axis) was surgically removed 2 weeks p.i. and the right pln (y-axis) 10 weeks p.i. The frequency of GC-Tfh clonotypes present in GCs of contralateral pln is shown as dot plot diagram at distinct time points. Each dot represents one GC-Tfh clonotype. Linear regression lines ranging from 1 to 10 and r2 values are depicted for each mouse (n = 5, mouse 1: black; mouse 2: red; mouse 3: green; mouse 4: blue; mouse 5: purple). (d) The 20 most frequent GC-Tfh clonotypes present in the left 2 weeks pln (black dots) are compared to their presence in the right 10 weeks pln (red dots). One representative comparison out of five is shown. (e) The relative intersection of the data in (d) was calculated (mean ± SD; **p<0.01, Mann–Whitney U-test, n = 5, two pln each). The online version of this article includes the following figure supplement(s) for figure 4:

**Figure supplement 1.** The synchronization of T cell clonotypes between contralateral lymph nodes begins 3 days after priming.

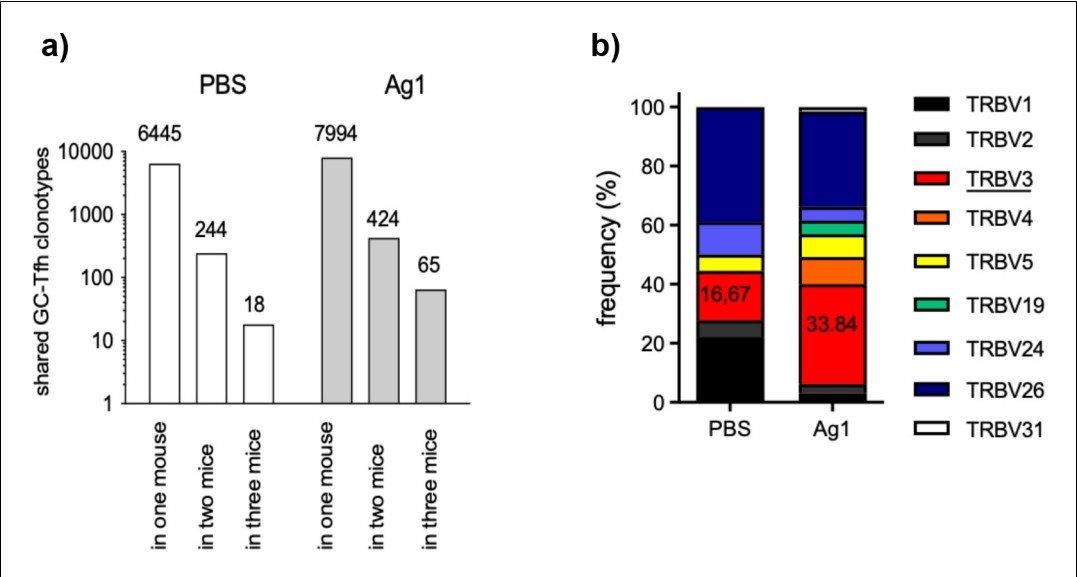

**Figure 5.** More GC-Tfh clonotypes are shared between the mice of the Ag group. (a) The number of GC-Tfh clonotypes of the phosphate-buffered saline (PBS) group and Ag1-group that were found in one mouse, in two, or in all three mice at 4 weeks post injection as described (*Madi et al., 2014*). (b) The relative distribution of TRBV segments in all mice is shown. Especially, TRBV3 segment (red) predominates in shared Tfh clonotypes.

## Discussion

CD4 T cells are pivotal in adaptive immune responses during infection, autoimmune diseases, cancer, and vaccinations. Each CD4 T cell expresses one specific αβ TCR interacting with specific peptide-Ag presented in MHCII complexes (*Itano and Jenkins, 2003*). Upon interaction of the TCR with peptide:MHCII complexes, naive CD4 T cells undergo proliferation and functional differentiation into distinct T helper subsets such as Th1, Th2, Th17 effector cells, Tfh, and immunosuppressive Treg cells (*Zhu et al., 2010*). The main factor that regulates CD4 T cell differentiation is the TCR signaling strength, which is controlled by TCR/peptide:MHCII interactions, co-stimulation, and/or optimal dwell times. The differentiation into Tfh depends on strong TCR signals. It is suggested that they are strictly Ag-specific (*Hwang et al., 2015*; *Knowlden and Sant, 2016*; *Tubo et al., 2013*; *Merkenschlager et al., 2021*). Consistently, it has been shown that the maintenance and expansion of Tfh requires sustained Ag stimulation (*Baumjohann et al., 2013*; *Merkenschlager et al., 2021*).

In this study, we asked how this particular CD4 T cell subset is distributed within one individual at the clonal level. We used an autoantibody-mediated disease model of the skin, epidermolysis bullosa acquisita, and induced GC-Tfh in two separate draining pln by injecting auto-Ag in adjuvant (Ag1 or Ag2 group) or adjuvant only (PBS group).

Our study shows two major findings. First, we found a high overlap of dominant GC-Tfh clonotypes in both pln of one mouse (*Figure 2a, b*). This data demonstrates that even though the response to the auto-Ag was polyclonal in each mouse, a group of almost identical GC-Tfh clonotypes became dominant in each draining pln. Thus, exposures to the auto-Ag (Ag1 or Ag2) decreased the diversity of the endogenous Tfh repertoire compared to the PBS group. To establish this finding, we used multiplex iRepertiore PCRs and subsequent MiTCR analysis for preparing libraries and annotating TCRβ sequences (*Bolotin et al., 2013*), which ensures to yield the highest possible diversity of TCRβ clonotypes (*Afzal et al., 2019*). To confirm this decrease in diversity of the endogenous Tfh repertoire, we reanalyzed our data for Ag1 with a more precise analysis tool MiXCR that is advantageous for error corrections and adjusts for a more accurate clonal composition (*Bolotin et al., 2013*; *Bolotin et al., 2015*; *Team I, 2019*). The high overlap of the dominant Tfh clonotypes in the Ag1 group was clearly confirmed (*Figure 2—figure supplement 1*). To further compare both analysis tools, we calculated the MHI from samples of the Ag1 and PBS groups analyzed either with MiTCR or MiXCR. Data show a high similarity in both the Ag1 group (0.70 ± 0.04) and the PBS group (0.69 ± 0.039). Obviously, our finding that Tfh clonotypes highly overlap in

contralateral pln is very robust independent of the analysis tools. On the other side, the suitability of the multiplex PCRs can be seen by the correct detections of the deletions in the TRBV genes of SJL mice (*Figure 3c*; *Behlke et al., 1986*). In summary, this data that the endogenous Tfh repertoire decreased after Ag exposure indicates that the number of T cell clones specific for the injected auto-Ag is restricted per mouse and supports previous reports demonstrating that the precursor frequency specific for auto-Ag within the T cell population of an individual is rather low (*Jenkins and Moon, 2012*). However, the precursor frequency of naive T cells specific for different peptide:MHC complexes varies in size. Therefore, it will be interesting whether this high clonal overlap would be found after injection of xenogeneic or pathogenic Ag. Moreover, because both Ag used in this study are polypeptides of a relatively large size (200–400 aa) and clearly differ from the size of peptides (13–25 aa) that are usually presented in MHCII (*Natarajan et al., 2018*), it is likely that they contain more than one antigenic epitope. It is reasonable to assume that injecting smaller peptides will further narrow the diversity of the individual GC-Tfh repertoire. In an initial experiment as an example for a recombinant xenogeneic protein antigen, we injected the *Schistosoma mansoni*-derived gluta-thione-s-transferase (GST)-tag of Ag1 (*Rao et al., 2003*), emulsified in TM into both footpads, and analyzed the distribution of Tfh clonotypes in both draining pln 4 weeks p.i. (*Figure 2—figure supplement 2*, *Supplementary file 6*). We found that the distribution of the GST-specific Tfh clonotypes between both pln was less controlled compared to the auto-Ag1. Dot plot diagrams showed a higher variability between the mice. In line, the MHI was not significantly different from the PBS group (*Figure 2—figure supplement 2*) and the number of overlapping Tfh clonotypes is lower in the GST group compared to the Ag1 group (*Figure 3—figure supplement 1*). Analysis of the V/J usage revealed significant differences in expression of TRBV3. The J2-4-segment was significantly less abundant after GST/TM immunization compared to the PBS/-group and the J2-5 segment was jointly higher abundant after GST/TM and Ag1 immunization compared to the control group (*Figure 3—figure supplement 1*). These data confirm the strict Ag specificity of Tfh cells in GCs, especially during the first 4 weeks p.i. The higher inconsistency in the number of overlapping Tfh clonotypes in the GST group could be explained by an earlier consumption of the Ag or the presence of more high-affinity T cell clones. These findings might be strongly influenced by the time point and the Ag concentration, which should be further investigated in future studies.

Another arising question is when this overlap of T cell clonotypes between both pln of one mouse starts. For example, T cells could travel between pln and the clonal selection could be a continuous organism-wide systemic process. The presence of circulating Tfh cells has been described in humans and mice (*Brenna et al., 2020*; *He et al., 2013*); however, it is not clear whether these circulating Tfh cells can enter ongoing GC reactions. Alternatively, it could take place during the priming phase. In this case, either the naive TCR repertoire is sufficiently broad that the Ag-driven selection for the GC reactions in both pln would robustly yield groups of similar clones independently or recently activated CD4 T cell clones would exchange between both pln after priming before GC reaction starts. To address this, we performed an initial experiment and identified TCRβ sequences of entire left and right pln of the same mouse (naive mice and mice that had been exposed to Ag1 for 1 or 3 days; please find detailed information in *Supplementary file 5*) and compared the relative percentage of overlapping T cell clonotypes between left and right pln by using the MHI. We found that the MHI did not differ between naive mice and Ag-exposed mice at 1 day p.i. (0.2035 + 0.01962 in naive mice, 0.2113 + 0.009 at 1 day p.i.), but a significant increased MHI at 3 days p.i. (0.3716 + 0.031; p<0.01, Kruskal–Wallis test, mean + SD for n = 3 mice, *Figure 4—figure supplement 1*). In parallel, the number of proliferating T cells and the mRNA expression for the proliferation marker Ki67 increased significantly (*Figure 4—figure supplement 1*). One may speculate that a high number of progenies emerges from the high-affinity T cell clones 3 days p.i. (*Cho et al., 2017*), which then migrate into other activated lymphoid tissues before differentiating into GC-Tfh. In this case, the initial progeny of activated T cell clones could originate from identical precursors, which would enter the circulation and migrate into the other pln, in which they outcompete low-affinity competitors and develop into GC-Tfh. However, on the other side it is assumed that Tfh stay resident in lymph nodes to provide B cell help (*Crotty, 2019*). One possibility to observe potential exchanges of Tfh cells between lymph nodes would be treatment with the T cell trafficking blocker FTY720. However, previous studies revealed no effect in this autoimmune mouse model (*Niebuhr et al., 2017*; *Thieme et al., 2019*). The second major finding of our study is that the PBS group, which received only adjuvant, develops GCs to a similar extent as the Ag group. Here, in contrast to the Ag group,

the GC-Tfh repertoire remains diverse. No overlap of dominant GC-Tfh clonotypes was found between both pln. TM is a water-in-oil adjuvant that contains squalene among other components but no peptides that could be presented in MHCII (TiterMax Inc). It is proposed that adjuvants stress or kill local cells, which leads to the release of endogenous peptide-epitopes that are subsequently presented to T cells (*Riteau et al., 2016*) and induce the formation of GC reactions as observed in this study (*Figure 1c*). In this scenario, numerous endogenous peptide-epitopes at low concentrations might be released by the local inflammation, which leads to the diverse GC-Tfh repertoire in both pln (*Figure 2a, b*). However, to break the tolerance in our model, both auto-Ag had to be injected at very high concentrations (60 µg/Ag1 or 120 µg/Ag2, respectively) (*Sitaru et al., 2006*). This data leads to the hypothesis that the concentration of the Ag is critically involved in the emergence of overlapping GC-Tfh repertoires in separate pln (summarized in *Figure 6*). It will be interesting to find out how low Ag concentrations at distinct tissue sites affect the GC-Tfh repertoires. Moreover, it will be important to study the adjuvant-stimulated peptide-epitopes in future studies,

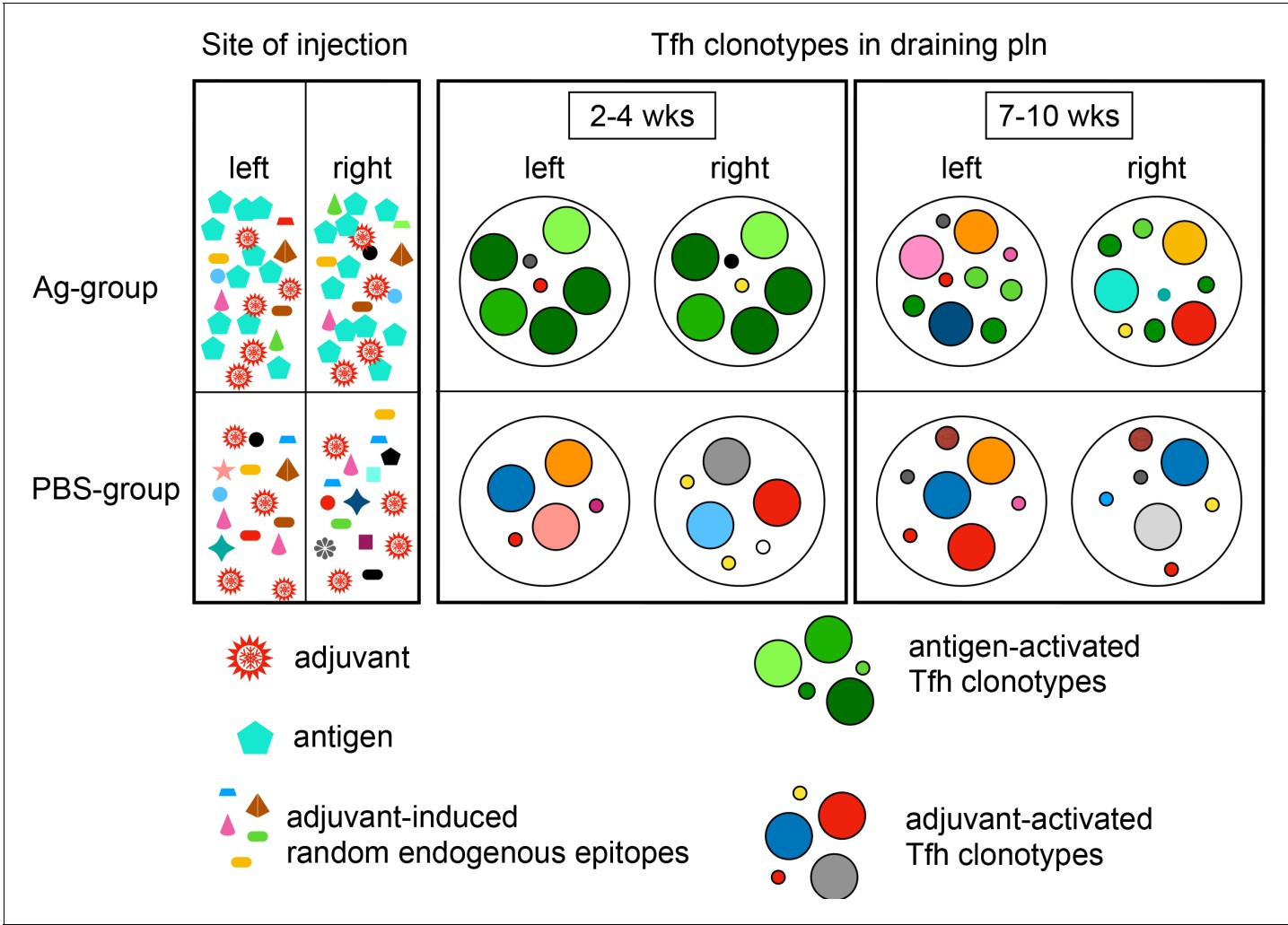

**Figure 6.** GC-Tfh clonotypes overlap in separate lymph nodes only after Ag exposure. Simplified drawing summarizes the distribution of GC-Tfh clonotypes of the Ag group and PBS group over time. Clonotypes are displayed as circles. Circle colors indicate specificity and circle size its frequency. Ag is injected at high quantities into both footpads (bluish green pentagons, top left). The injection of adjuvant induces a local inflammation and the release of many endogenous epitopes at low quantities (diverse forms and colors, bottom left). Both groups receive adjuvant (red jagged circles). Identical high-expanded Ag-activated Tfh clonotypes emerge in germinal centers (GCs) of both draining popliteal lymph nodes (pln) after 2–4 weeks (big greenish circles, top middle). The high-expanded Tfh clonotypes get exchanged after 7–10 weeks. The initially high-activated Ag-specific Tfh clonotypes remain at low quantities (small greenish circles, top right). Adjuvant-only-exposed pln harbor distinct random Tfh clonotypes (diverse colors, bottom middle and bottom right).

especially because squalene compounds are frequently used in human vaccines and might evoke unwanted autoimmune responses (*Pellegrini et al., 2009*).

Another conclusion of our study is that the total number of GC-Tfh within one individual is regulated by the number of lymphoid tissues that drain Ag-exposed tissue sites. This data suggests that Ag injection at two or more tissue sites could improve the development of high-affinity antibodies. We believe that this information will be highly useful for the development of new and more efficient vaccination strategies.

Finally, we wish to highlight that this in vivo approach could help to identify new peptide-specific TCR sequences and confirm already established datasets (*Teraguchi et al., 2020*). Even though TCR datasets are growing, the number of TCRs with known Ag specificity and function is extremely low (*Shugay et al., 2018*; *Zvyagin et al., 2020*). Especially the detection of self-reactive CD4 TCR is problematic due to the low binding affinities between TCR and autoantigenic-peptide:MHCII complexes (*Dolton et al., 2018*; *Rius et al., 2018*). Thus, even though our approach is restricted to mouse models, the use of HLA-transgenic mice might enable to link TCR sequences also to human antigenic epitopes.

# Materials and methods

## Mice

8–12-week-old female SJL/J mice were obtained from Charles River Laboratories (Sulzfeld, Germany), and 8–12-week-old female H2s-congenic C57BL/6 mice (B6.SJL-H2s) C3c/1CyJ (B6.s) mice were kindly provided by the Lübeck Institute of Experimental Dermatology (LIED, University of Lübeck, Lübeck, Germany). All experiments were performed at the animal facility of the University of Lübeck and approved by Animal Care and Use Committee of the state Schleswig-Holstein (Ministerium für Energiewende, Landwirtschaft, Umwelt, Natur und Digitalisierung), proposals: V242-45884/2016 (90-7/16), V242-7224.122-1 (35-3/12), V312-72241.122-1 (106-10), V312-72241.122-1 (104-10), V312-72241.122-1 (92-7/09), and 23/A11/05. All animal experiments were conducted by certified personnel.

## Induction of Tfh, GCs, and skin lesion

GCs and skin lesions were induced as described for the mouse model of experimental epidermolysis bullosa acquisita (*Hammers et al., 2011*; *Iwata et al., 2013*). Briefly, mCOL7c-GST (Ag1, 421 aa) is recombinantly produced and contains the subdomain c of the noncollagenous NC1 domain of mCOL7 (210 aa, 757–967) and is linked to the GST-tag (211 aa). Ag2 contains the VWFA2 subdomain of the noncollagenous NC1 domain of mCOL7, has a size of 190aa (1048–1238) and no GST-tag. mCOL7c (60 µg in PBS, Ag1 group), vWFA2 (120 µg in PBS, Ag2 group), GST (32 µg in PBS), or PBS (PBS group) without antigen were emulsified 1:1 in TM (HiSS Diagnostics GmbH, Freiburg, Germany) and injected in a volume of 60 µl s.c. in both hind footpads of SJL mice or MHCII-H2s-congenic C57BL/6 mice (*Iwata et al., 2013*). The inner sides of the ears were slightly scratched 1 week before analysis to obtain standardized skin lesions (*Niebuhr et al., 2020*). Mice were sacrificed 2, 4, and 7 weeks p.i. Pln and ear skin lesions were removed, snap-frozen, and stored at −80°C. To compare GC-Tfh repertoires within one mouse at distinct time points, one pln was surgically removed 2 weeks p.i. and the other at the end of the observation period after 10 weeks (*Ellebrecht et al., 2016*).

## Histological analysis, quantifications, and qRT-PCR

Serial cryosections of pln (10 µm thick for histology, 12 µm thick for laser microdissection) were mounted on plain glass slides for histology or on membrane-covered slides (Palm Membrane Slides, PEN membrane, 1 mm; Carl Zeiss AG, Germany) for laser microdissection (*Stamm et al., 2013*). GCs and GC-Tfh were identified by staining for proliferating cells with rat anti-mouse Ki67 mAb (BioLegend, Koblenz, Germany) and biotinylated rabbit anti-rat IgG (Dako, Glostrup, Denmark), or biotinylated mAbs against TCRβ and/or B220 (both from BD Biosciences) and visualized as described (*Stamm et al., 2013*; *Fähnrich et al., 2018*). Quantification of GC area and Tfh was performed with a standardized ImageJ pipeline. The cumulative area of the complete B cell zone of one cryosection was determined. Likewise, the cumulative area of all GCss of this cryosection was determined. The

relative proportion of the GC area and the B cell zone area was calculated as quotient. For each sample, three individual cryosections were evaluated and the mean proportion was used for evaluation. Tfh were identified from immunohistochemical staining for T-, B-, and proliferating cells with the function 'Colour Deconvolution.' The frequency of Tfh was determined in GCs via the function 'Analyze Particles.' The frequency of Tfh per volume was calculated as the product of the frequency per area times the thickness of the section (12 μm). Additional methods are described in *Supplementary file 7*.

### 3D reconstruction

A complete collection of 14 μm serial cryosections (approximately 200 sections per pln) from an entire pln was stained for T-, B-, and proliferating cells (as described above) and imaged with an automatic slide scanner (Panoramic SCAN II; 3D Histech) and processed by For3D. ImageJ and homemade MATLAB functions were used to render pln sections into 3D. GCs were segmented by filtering, thresholding, and soothing the stack of pln section images. MATLAB was used to identify individual volumes of the 3D-GC structures within the pln as described (*Fähnrich et al., 2018*; *Irla et al., 2013*). The GC volume distribution was assessed, and the GC numbers in both pln were estimated according to their characteristic size of approximately $5 \times 10^6$ μm³ as described (*Wittenbrink et al., 2010*).

### Isolation of GC-Tfh by laser microdissection

GC-Tfh were obtained by carefully laser-capturing entire GCs including the light zones with accumulating Tfh with the PALM MicroBeam laser microdissection system (Carl Zeiss AG, Oberkochen, Germany) (*Stamm et al., 2013*). To estimate the GC volumes, the isolated GC areas were determined by the PALM MicroBeam software (Carl Zeiss AG) and multiplied by the section thickness (12 μm). Thereby, it was aimed to extract a volume of on average $40 \times 10^6$ μm³ (*Supplementary files 1* and *2*).

### Flow cytometry and antibodies

A total cell number of $10^6$ single cells from each (left and right) pln was stained using APC-conjugated anti-mouse CD4 (Clone GK1.5, BioLegend, San Diego, USA), BV510-conjugated anti-mouse CD8a (clone 53-6.7, BioLegend), PerCPCy 5.5-conjugated anti-mouse CD45R/B220 (clone RA3-6B2, BioLegend), PE/Cy7-conjugated anti-mouse CD185/CXCR5 (clone L138D7, BioLegend), and BV421-conjugated anti-mouse CD279/PD-1 (clone 29.F1A12, BioLegend). Samples were analyzed on a BD Biosciences LSRIII flow cytometer and Tfh were identified as CD4/CXCR5/PD-1 co-expressing cells (*Meli and King, 2015*). $5 \times 10^4$ Tfh were sorted and the repertoire of TCRβ clonotypes was identified as described above (*Supplementary file 3*).

### Identification of TCRβ clonotypes by Illumina Miseq sequencing

For identification of TCRβ sequences, laser-captured GC, 40 serial skin cryosections or $5 \times 10^4$ CD4+/CXCR5+/PD-1+ T cells were used for RNA isolation as described above. The preparation of cDNA and amplification of the Ag-binding site (CDR3β region) of the TCRβ chains were performed according to the manufacturer's protocol (iRepertoire, patent 7999092, 2011, Huntsville, USA) and prepared for pair-end sequencing with the Illumina Miseq system as described (*Li et al., 2017*). CDR3 identification, clonotype clustering, and correction of PCR and sequencing errors were performed using ClonoCalc wrapping MiTCR software according to the IMGT nomenclature (*Bolotin et al., 2013*; *Fähnrich et al., 2017*; *Lefranc et al., 1995*). Only annotated TCRβ clonotypes, defined as in-frame TCRβ clonotypes with a copy number ≥2, were further considered (*Madi et al., 2014*). Additionally, to avoid any artificial diversity, which could originate from unpredictable PCR errors or unrelated bystander T-lymphocytes, only annotated TCRβ clonotypes with a relative abundance above the median were kept and evaluated. Analysis of the TCRβ repertoire was realized using the R programming language and was based on the tcR package (*Nazarov et al., 2015*). Some data were additionally analyzed using MiXCR and the web-based tool Immunarch (*Bolotin et al., 2015*; *Team I, 2019*). Data for GC-Tfh or for skin sections are summarized in *Supplementary files 1–6* and published (*Niebuhr et al., 2020*).

## Statistical analysis

Statistical analyses were performed using the R programming language or GraphPad Prism 5.0 (GraphPad Software Inc, La Jolla, USA). Statistical significance was assessed by Mann–Whitney U-test, and multiple comparisons were performed using Kruskal–Wallis test or two-way ANOVA with Sidak's correction test (n = 3, two pln each). A p value of <0.05 was considered statistically significant.

## Acknowledgements

We thank L Gutjahr, P Lau, R Pagel, and D Rieck for their technical assistance. This study was funded by the German Research Foundation (DFG) within the framework of the Schleswig-Holstein Excellence Cluster I and I (EXC 306, Inflammation at Interfaces, project XTP4), the graduate school GRK 1727/2 and the TR-SFB654 project C4 at the University of Lübeck. We acknowledge financial support by the Land Schleswig-Holstein within the funding program Open Access Publikationsfonds.

## Additional information

### Funding

| Funder | Grant reference number | Author |
|---|---|---|
| Deutsche Forschungsgemeinschaft | EXC 306 | Kathrin Kalies |
| Deutsche Forschungsgemeinschaft | GRK 1727/2 | Jürgen Westermann<br>Kathrin Kalies |
| Deutsche Forschungsgemeinschaft | TR-SFB654 | Jürgen Westermann<br>Kathrin Kalies |

The funders had no role in study design, data collection and interpretation, or the decision to submit the work for publication.

### Author contributions

Markus Niebuhr, Data curation, Software, Formal analysis, Validation, Investigation, Methodology; Julia Belde, Christoph M Hammers, Katja Bieber, Data curation, Investigation; Anke Fähnrich, Data curation, Formal analysis, Validation, Methodology; Arnauld Serge, Christoph T Ellebrecht, Data curation, Methodology; Magali Irla, Data curation, Investigation, Methodology; Jürgen Westermann, Resources, Funding acquisition; Kathrin Kalies, Conceptualization, Supervision, Funding acquisition, Investigation, Methodology, Writing - original draft, Project administration, Writing - review and editing

### Author ORCIDs

Anke Fähnrich  http://orcid.org/0000-0003-1904-9544
Arnauld Serge  http://orcid.org/0000-0002-1280-6277
Magali Irla  http://orcid.org/0000-0001-8803-9708
Christoph M Hammers  http://orcid.org/0000-0001-5631-2415
Katja Bieber  http://orcid.org/0000-0002-3855-6683
Kathrin Kalies  https://orcid.org/0000-0002-8929-4249

### Ethics

Animal experimentation: This study was performed in strict accordance with the recommendations in the Guide for the Animal Care and Use Committee of the state Schleswig-Holstein (Ministerium für Energiewende, Landwirtschaft, Umwelt, Natur und Digitalisierung), proposals: V312-72241.122-1 (19-2/08), V312-72241.122-1 (92-7/09), V 313-72241.122 (92-7/09), V 312-72241.122-1 (104-10), V 312-72241.122-1 (106-10), V242-7224.122-1 (35-3/12) and 23/A11/05. All animal experiments were conducted by certified personnel.

Decision letter and Author response
Decision letter https://doi.org/10.7554/eLife.70053.sa1
Author response https://doi.org/10.7554/eLife.70053.sa2

## Additional files

### Supplementary files

• Supplementary file 1. Table displays volumes, T cell numbers, raw reads, and total and unique TCRβ sequences obtained from laser-captured germinal centers of popliteal lymph nodes (Ag1/SJLH2s).

• Supplementary file 2. Table displays skin lesion sizes, T cell numbers, raw reads, and total and unique TCRβ sequences (Ag1/SJLH2s).

• Supplementary file 3. Table displays T cell numbers, raw reads, and total and unique TCRβ sequences (Ag1/SJLH2s) from Tfh isolated by flow cytometry.

• Supplementary file 4. Table displays volumes, T cell numbers, raw reads, and total and unique TCRβ sequences from laser-captured germinal centers of popliteal lymph nodes (Ag2/C57BL6-H2s).

• Supplementary file 5. Table displays raw reads, and total and unique TCRβ sequences of entire popliteal lymph nodes (Ag1/SJLH2s).

• Supplementary file 6. Table displays volumes, T cell numbers, raw reads, and total and unique TCRβ sequences from laser-captured germinal centers of popliteal lymph nodes (GST/SJL-H2s).

• Supplementary file 7. Supplementary material and methods for *Figure 1—figure supplement 1* and *Figure 4—figure supplement 1*.

• Transparent reporting form

### Data availability

T cell receptor-RNA sequencing data generated in this study are deposited in the sequence read archive hosted at https://www.ncbi.nlm.nih.gov/sra under the primary accession code PRJNA731654. Some of the data (skin effector T cells) have been deposited under the accession code PRJNA586880.

The following datasets were generated:

| Author(s) | Year | Dataset title | Dataset URL | Database and Identifier |
|---|---|---|---|---|
| Kalies K, Niebuhr M | 2021 | TRBV sequences of murine follicular T helper cells in experimental epidermolysis bullosa acquisita | https://www.ncbi.nlm.nih.gov/bioproject/?term=PRJNA731654 | NCBI BioProject, PRJNA731654 |
| Kalies K, Niebuhr M | 2021 | TCRb-seq data from dermal T cells in SJL mice upon immunization | https://www.ncbi.nlm.nih.gov/bioproject/?term=PRJNA586880 | NCBI BioProject, PRJNA586880 |

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
