## [Decision Letter]

[Editors' note: this paper was reviewed by Review Commons.]

**Acceptance summary:**

This paper makes use of an elegant and technically complex approach to study the T cell receptor (TCR) clonotype dynamics of follicular helper T (Tfh) cell during an auto-antigen challenge, which is done by laser capturing of germinal centres inside popliteal lymph nodes, and combined with TCR sequencing, clearly isolating differentiated Tfh cells.

---

## [Author Response]

Reviewer #1 (Evidence, reproducibility and clarity (Required)):Niebuhr et al. use a mouse model of the autoimmune disease epidermolysis bullosa acquisita to investigate the T cell receptor repertoire of follicular helper T cells (TFH) isolated from the light zone of germinal centers by laser capture as determined using high throughput sequencing. The key question the authors address is whether such repertoires differ when the same mouse is immunized in two different locations in parallel. Two weeks after immunization dominant T cell clones were comparable in germinal centers in the two lymph nodes that drain the two different injection sites. The size of such clones was contracting at 7 and 10 weeks after injection and similarities of repertoires between the two lymph nodes were lost. These are well executed complex experiments. Welcome are the repertoire characterization of the PBS control, the comparison with effector T cells, and the use of a second mouse model for corroboration. The data support the idea that the immune response to the autoantigens is comparable across the entire mouse rather than distinct in each draining lymph node.

We thank the reviewer for this valuable note.

The compact manuscript lacks causal experiments. The authors could for example prevent lymphocyte trafficking between lymph nodes to distinguish whether the similar TCR repertoires in the two draining lymph nodes two weeks after injection arise from independent parallel priming or from lymphocyte trafficking between the lymph nodes. Such experiments would substantially strengthen the manuscript.

Thank you for raising this point. We agree with the reviewer that is important to examine lymphocyte trafficking. We surgically removed one 2wk lymph node and compared Tfh clonotypes to the 10wk contralateral lymph node of the same mouse, which reflects an inhibition of T cell trafficking between both lymph nodes for a period of 8 wks. The data show that especially the size of the dominant 2wk Tfh clonotypes decreased in the 10 wk lymph nodes and other Tfh clonotypes became dominant but most of the 2 wk Tfh clonotypes were maintained within the GC over this period of 8 wk. We agree with the reviewer that the changes observed indicate an exchange of dominant clones within one lymph node, but it cannot be concluded that Tfh clonotypes exchange between lymph nodes. To clarify this, we rewrote the manuscript substantially (title, abstract, second paragraph on page 7, line 166 and 181).

In addition, we performed experiments and compared T cell receptor sequences in left and right lymph nodes of the same mouse 1d and 3d after priming. The number of shared dominant T cell clonotypes increased 3d p.i.. (Morisita Horn index). We included these data in the discussion (page 11, line 267).

Looking at the week 7 and 10 TFH TCR repertoires, Niebuhr et al. interpret the lack of significant correlation between the repertoires in the two draining lymph nodes as evidence for clonotype replacement (line 157). There may be a simpler explanation that should be considered. Clone sizes contract over the course of the immune response as seems evident in Figure 4A. This should be quantified. As the repertoire of non-antigen-specific follicular helper T cells is different in each lymph node, a contraction of the antigen-specific clones to the size of the non-antigen-specific ones may simply make it technically impossible to follow the antigen-specific clones on the variable background of the non-antigen-specific ones. In this context, the linear relation analysis used by the authors is largely driven by the relatively small number of large clones.

Thank you for raising this basic concern. We agree that our data focus especially on the high frequent clones that are shared between left and right lymph nodes. In our opinion these are the most relevant Tfh clonotypes to look at due to the limited size in GC and the high clonal competition between Tfh for space within GC (Merkenschlager et al. Nature 2021, doi: 10.1038/s41586-021-03187-x). Our data show that the size of the overlapping dominant 2 wk Tfh clones declines over time and the size of other potential bystander activated Tfh clones becomes superior (Figure 4). This does not mean that the initial large clones would be completely replaced. Instead, they are still there but at lower frequencies. We rewrote the manuscript accordingly (title, abstract, second paragraph on page 7, line 166 and 181). In addition, to study the clone distribution we performed box plot analysis from the 2wks and 10wks GC-Tfh clonotypes (Figure 4c and d) that shows that the overall clone size of GC-Tfh does not generally contract over time.

As minor comments, statistics for Figure 1C should be given. Arrows in Figure 1D may be shifted.

We appreciate this comment and added statistics and changed the arrows

accordingly.

Reviewer #1 (Significance (Required)):The manuscript leaves important questions unresolved.A similar TCR repertoire in the distinct lymph nodes draining two injection sites could in principle be caused by two different mechanisms. The naïve T cell repertoire in each lymph node could be sufficiently broad that the antigen-driven selection in the germinal centers robustly yields independent groups of similar clones. Alternatively, lymphocytes could travel between lymph nodes thus setting up organism-wide clonal selection. These scenarios can be experimentally distinguished as discussed above.

Thank you for raising this point. The surgical removal of one lymph node resembles an inhibition of T cell trafficking within one mouse. In addition, we performed new experiments that show the synchronization between both lymph nodes starts during initial T cell proliferation 3d p.i.. However, we cannot conclude whether these shared T cell clonotypes would differentiate into GC-Tfh cells. Even though the idea to inhibit lymphocyte trafficking is appealing we feel that this kind of kinetic analysis that would be the required is beyond the scope of this paper. We rewrote the manuscript substantially (title, abstract, second paragraph on page 7, line 166 and 181).

As is, the manuscript describes that TFH TCR repertoires are similar in different draining lymph nodes. Are there any practical consequences to that? There should not be, one injection should be sufficient. Are there any practical consequences to the mechanism ensuring this similarity, as outlined in the first question? If there are, this should be discussed.

Thank you for asking these clarifying questions. To better emphasis the practical consequences of our manuscript we changed the focus towards Tfh clonotypes during the course of an autoimmune disease and the importance of timing. Usually in autoimmune diseases, multiple tissues sites are affected, and it has been suggested that autoreactive clonotypes accumulate in the most adjacent lymphoid organ, which was simple judged by V/J gene segment expression (Oftedahl et al. 2017, doi: 10.1016/j.jaut.2017.03.002). In contrast to this, another report showed that high numbers of bystander-activated Tfh clonotypes accumulate also in autoimmune models (Ritvo et al., 2018, doi: 10.1073/pnas.1808594115). Our data fills the gap by showing that indeed identical Tfh clonotypes accumulate in GC of lymph nodes that drain autoantigen-exposed skin sites but that this high overlap of prominent Tfh clonotypes is only transiently. In addition, our approach has the potential to identify antigen-specific T cell clones in vivo and could help to improve vaccination strategies. To emphasize these practical consequences we rewrote the manuscript profoundly (title, abstract, main text and discussion).

The key expertise of this reviewer is in T cell signal transduction. This review thus presents a more general immunological view from outside of the core question of TFH TCR repertoires in the germinal centers.Referees cross-commentingI share the conceptual concerns of reviewer 2. I regard laser capture of individual germinal centers and whole lymph node cytometry as complimentary, each having its strengths and weaknesses. I can't comment of the choice of sequencing approach as discussed by reviewer 3.Reviewer #2 (Evidence, reproducibility and clarity (Required)):The manuscript by Niebuhr et al. used laser dissection and TCR sequencing to address the role of follicular T helper (Tfh) cell clonality after immunization in mice. While the topic of the manuscript is of interest, the experimental data and several technical concerns do not fully support the conclusions drawn from the study.Major points:1. Mice were immunized s.c. in both hind foot pads with either a model antigen and Titermax adjuvant or with PBS and Titermax adjuvant, i.e. one mouse received either the Ag or the PBS control. They found that 90% of the clones overlapped in the LNs and that this was transient. The authors claim that TCRs failed to remain enriched and conclude from that the selection of germinal center (GC)-Tfh is controlled by systemic clonal competition throughout the response as stated in the abstract.The connection between the enrichment of clones (which is induced by the expansion of activated T cells, most likely clonal, which is not surprising) and its control by "systemic clonal competition" remains questionable though.

We thank the reviewer for drawing our attention to the lack of clarity. We agree with the reviewer that an accumulation of highly expanded T cell clones after immunizations is not surprising. However, we showed previously that high expanded T cell clones even within adjacent T cell zones of murine spleen do not substantially overlap after antigen exposure (Textor et al. doi: 10.4049/jimmunol.1800091). Therefore, the huge difference in the number of shared Tfh clones in both lymph nodes between the Ag and PBS group is surprising. We used the term “systemic versus local selection” to explain this difference between the PBS and Ag group. The adjuvant in the PBS group induces a local inflammation in both footpads leading to the presentation of multiple local epitopes. Each lymph node develops its local Tfh repertoire that does not overlap with the other lymph node. In contrast, the Tfh repertoire of both lymph nodes overlaps in the Ag group. We called this “systemic” because both lymph nodes are involved. We understand that this wording might be misleading. To clarify this, we rewrote the abstract, and changed it throughout the text.

Conceptually, one big question remains: Why did the authors not use both Ag challenge and control treatment in the same mouse? The foot pad/popl. LNs are ideally suited to test this in one mouse. The presented data further do not allow the conclusion that "the GC response is shared between separate LNs and underlies a constant systemic competition" (line 270/271 and similarly in the abstract). Another point is the timing of 2,4, and 7 weeks, which appears quite late given that GCs (admittingly dependent on the Ag/adjuvant used) often reach their peak magnitude already around 7-14 days post immunization.

Thank you for suggesting this experiment. This kind of experiment should be clearly done. It would be also interesting to inject different antigens and repeat the injections. However, even though it would substantially add interesting information, we feel that this would be beyond the scope of this manuscript and would extend it too much. Additionally, we rewrote substantial parts of the manuscript and avoided terms like “shared GC responses and systemic competition”. Regarding the timing, the time points were chosen depending on the onset of skin pathologies. It takes at least 4 wks for the development of initial skin lesions. This long period is probably required for the loss of tolerance and the recruitment of autoreactive T cell clones. To address earlier time points, we compared the T cell repertoire of complete lymph nodes 1d p.i. and 3d p.i..(discussion page 11, line 267, supplemental Figure 2).

2. The approach of dissecting GCs with laser capture is appealing, however, it would be much easier to use sorting by flow cytometry, as this would not only allow single cell sequencing analysis (which was not performed here, only bulk in Figure 2.) but flow cytometry would also provide a much more powerful analysis of the Tfh cells in this setting. While the authors argue that the observed skin phenotype is depending on high-affinity Abs derived from GCs, GC B cells were not assessed here at all, even though a B cell read-out would further help in dissecting the Tfh differences between Ag/TM vs. PBS/TM conditions.

We appreciate this comment. In this approach we aimed to analyze Tfh in GC without disturbing the organized structure of the lymph nodes. This approach has the advantage that Tfh cell can be analyzed in vivo, which avoids any bias such as unwanted T cell activations or deaths during cell isolation pro. Flow cytometry was used as control. Interestingly, the similarity was higher in FACS-sorted GC-Tfh clonotypes compared to lasercaptured GC-Tfh clonotypes (MHI, 0.46 ± 0.07 versus 0.71 ± 0.11, mean ± SD, Figure 2f). This data indicates that each GC within one lymph node shares the majority of Tfh clonotypes. We included this result in the text (page 6, line 137). We agree with the reviewer that FACS should be the method of choice for further studies. We have not expected before that lasercaptured GC and FACS sorting would give such similar results. Regarding B cell data, we feel that this kind of new experiments would be beyond the scope of this manuscript.

3. Principally, sorting works (Figure 2e-h). Nevertheless, there are some concerns here: A) The gates in Figure 2e appear (hopefully) not correctly positioned. Did it shift upward during the preparation of the figure? B) From the plot and the legend, it seems that all CXCR5+PD-1+ cells were sorted. These cannot be regarded as GC-Tfh cells, since GC-Tfh cells are only those cells with very high expression of both markers.

Thank you very much pointing towards this important issue. We apologize for the

bad quality of this FACS plot. We now added the complete gating strategy (Figure 2e).

4. The authors use very low numbers of replicates for their experiments. While complex assays such as laser capture and subsequent sequencing require a lot of effort, it needs to be made sure that the results are robust and reproducible. Claiming in Figure 1b that there is no difference between left and right pLN if there are only two mice shown and in mouse 2 there are ~35 vs. ~55 GCs in left vs. right LN, underscores this (with n=2 being not enough for statistical use). Same for 1c.

Thank you! We understand the concern of the reviewer, changed Figure 1b and

1c and rewrote this sentences (page 5, lines 97 and 113). It is more important that GC reactions emerge in both pln instead that the number of GC are identical.

5. Lines 117-119: “This data demonstrate that distinct T cell clones contribute to the responses in GC and skins, which is unexpected considering the fact that all T cells were activated with the same Ag.” I would argue that this is actually expected, since those cells that emigrate are most likely not the same cells that end up in GCs, i.e. they are potentially primed at different sites within the T zone or at the T-B border, with different signal strengths, and additionally at different time points (Marc Jenkins’ work). Many variables here. Furthermore, mice were immunized with complex antigens most likely containing several different epitopes.

We appreciate this comment and apologize for this inconclusive conclusion. We

agree with the reviewer and rewrote this paragraph (page 6, lines 133).

6. It appears that during cell sorting by flow cytometry dead cells were not excluded with a viability dye. That should always be done to reduce background/false-positive staining, and is regarded as good scientific practice. This is particularly important for sensitive down-stream applications such as sequencing.

To clarify this issue we included the complete gating strategy (Figure 2e).

7. The authors injected up to 120µg of Ag in 60µl total volume in the hind foot pads. That is in both regards a lot. Is this due to the autoantigen-driven immunization model that would otherwise not precipitate a phenotype? Is the observed data dependent on such high antigen-load? Would the results be different if other exogenous (model)antigens would be used at lower concentrations?

Thank you for raising this interesting questions. This high dosage of antigen is required to induce pathologies in this skin blistering autoimmune model. It will be important in further studies to use different kind of antigens and antigenic peptides.

Minor points8. Some words should be revised, e.g. "skins" line 118, "maintenances" line 144, "Therefore" line 162, "avoid" line 262, "scarified" line 312.

Thank you. We revised all points accordingly.

9. Line 335, provide a number or rough estimate of the serial sections used for the reconstruction.

Thank you. We included this information on page 15, line 367.

10. What is the time point of the PBS/TM data in Figure 5? Also week 2?

Thank you. The time point for PBS/TM is 4 wks p.i.. To clarify this, we changed Figure 5 and rewrote the respective paragraph (page 8, line 204).

Reviewer #2 (Significance (Required)):The question of clonality of endogenous Tfh cell response is relevant and open questions remain. The antigen-specificity in this system is not entirely clear, since complex antigens were used. The clonality of the Tfh cells response has been adressed in humans before, e.g. in influenza infection: Brenna et al. Cell Rep 2020 (PMID: 31914381) found that TCR sequences showed similarities between tonsils and the periphery. This study shouldbe be discussed.

Thank you for these comments. We agree with the reviewer that peptides should be used for further analysis. Regarding the antigen-specificity recent data demonstrated that GC-Tfh cells are selected by affinity and TCR receptor signaling strength and underly a high clonal competition for space within GC (Merkenschlager et al. Nature 2021, doi: 10.1038/s41586-021-03187-x). Therefore, especially the dominant Tfh clones should be antigen-specific. In addition, the study of Brenna et al. shows that also circulating Tfh cells are antigen-specific and overlap with GC-Tfh clones in tonsils. Both studies were included into the introduction and discussion (page 3, line 59; page 10, line 231 and page 11, line 262)

Reviewer #3 (Evidence, reproducibility and clarity (Required)):The article "Receptor repertoires of murine follicular T helper cells reveal shared responses in separate lymph nodes" summarizes the results of a beautiful and technically complex study devoted to the Tfh clonotype dynamics in antigenic response. The general design of the experiment is fascinating. Authors managed to perform both short-term and long-term experiments on littermate mice, performed beautiful work of laser capturing of germinal centers inside popliteal lymph nodes, and combined the technology with TCR sequencing, clearly isolating differentiated Tfh cells. The reviewer would access this study from the point of view of TCR sequencing expert.

Thank you very much for this comment.

Major comments1. TCR sequencing data analysis is out-of date and not sufficiently justified. Instead ofMiTCR software more recent and customizable MiXCR software should be used. MiTCR was well-known to produce overestimation of diversity in samples, which is detrimental in conjunction with multiplex PCR approach of iRepertoire libraries. At the same time, MiXCR could be optimized for murine samples and multiplex PCR-based DNA libraries and provide adequate error correction. Data engineering is extremely important to check for overamplification in iRepertoire data, filter the cross-contamination between adjacent samples and avoid batch effects such as extensive clonal overlaps in top clonotypes in samples which were sequenced together on MiSeq. Authors should consider re-analysis of raw data with additional attention to batch effects as just considering clonotypes with at least 2 reads as real could be not enough to get a clean dataset. Usually with iRepertoire data you need to look at the suspicious expanded clones in sorted naïve T cells, which represent contamination from adjacent memory samples. With technical and biological replicates you can estimate the level of contamination and discard all clones with this threshold.

We appreciate these critical comments about our TCR analysis approach. However, many new TCR analysis approaches have been developed and computational comparative studies identified MiTCR as solid choice (Azfal et al. 2019, doi:

10.1093/bib/bbx111). Additionally, the higher diversity obtained in multiplex PCR enrichmentbased libraries makes sequencing easier compared to usage of 5’RACE based libraries. In this approach it is wanted and it is not problematic to yield high numbers of T cell clonotypes because:

(1) Due to the number of Tfh cells within GC, the number of Tfh clonotypes is considerable low when compared to entire lymph nodes (see table S1 and table S4). Each GC reaction emerged just in response to Ag or to adjuvant. This excludes the presence of potential false expanded memory T cells among naïve T cells.

(2) We focused only on dominant Tfh clones. All those TCR sequences below the median of the respective sample were removed. Thus, all clonotypes with 2 reads only do not play a role at all.

(3) Our data does not focus on the diversity within one sample instead we compared dominant Tfh clonotypes at two tissue sites. In our opinion, using MiTCR and iRepertoire rather strengthen our data due to the high yields of TCR sequences and our finding that the overall diversity decreases in the Ag exposed groups. In addition, a comparative study of computational methods for T-cell receptor sequencing data revealed that MiTCR and MiXCR do not profoundly differ in clonality measures (Azfal et al. 2019, doi: 10.1093/bib/bbx111). To address this point, we included this reference and wrote a paragraph in the discussion (page 10, line 235).

2. The manuscript contains inconsistent terminology, for example, TRBV segment naming. Axes on plots either lack labels or are incomprehensible in many cases. Therefore, the manuscript needs some editing.

Thank you for this comment! We edited the manuscript accordingly.

3. Authors describe that the time points in this study were selected accordingly to the development of experimental autoimmune process, the dynamics of its phenotype and its pathology – e.g. autoantibody deposition at the epidermis-dermis junction. However, it would be interesting to explore earlier processes of Tfh differentiation, competition for the antigen and establishment of the new GC. Do authors have the data on Tfh TCR dynamics in time points 4-14 days post injection of the antigen? The clonal competition could be more apparent and show if lateral lymph nodes develop the response independently. It may happen that in the first days the competition is restricted to lymph node and therefore private, and later the competition is systemic and both contralateral LN exchange large numbers of T cell clones. However this requires a whole set of experiments and may be out of scope of this particular short manuscript.

Thank you very much for these appealing suggestions. We performed new experiments and included the TCR analysis of contralateral draining lymph nodes 1d and 3d after priming. We found that the number of dominant T cell clonotypes that are shared between both lymph nodes increased 3d p.i.. (Morisita-Horn-index, Supplemental Figure S2). However, before this at 1d p.i. the number of shared clones between both pln decreased significantly (Jaccard Index, naïve: 0.082 ± 0.02, 1d: 0.053 ± 0.015 (p<0.05) and 3d 0.12 ± 0.01 (p<0.01), n = 3, 2 pln each, Mann-Whitney-U test). It might be that this decrease reflects the recruitment of new naïve T cells into the Ag-exposed lymph nodes before the Agactivated clones start to expand (Figure S2a). Subsequently, those clones that bound with highest affinity and therefore produced the highest number of progenies could distribute between both pln before they differentiate into Tfh cells. We did not include this data into the manuscript because further experiments will be required to answer this question.

Minor comments1. Authors need to choose 1 strategy of naming and keep it consistent. Mice groups are named Ag and control group, Ag and PBS group and, finally, + and – groups in Figure 3C, while meaning the same groups.

Thank you! We changed our manuscript accordingly.

2. TRBV segment names are used in 3 different ways. Authors need to choose 1 strategy of naming and keep it consistent, for example, keep IMGT recommended gene segment names.

Thank you! We changed our manuscript accordingly.

3. Figure 2A should have clonotype frequency annotated on axis label, r2 coefficient labeled on plot. Figure 3A, 4A, 4C – same comments.

Thank you! We changed the legends accordingly.

4. Figure 2C should have axes labeled. Effector T cells – why are they derived from ear? Text lacks the explanation; it only could be found deep in methods section. Labeling of axes should be more consistent.

Thank you! We changed the legends accordingly and included an explanation into the manuscript on page 6, line 124.

5. Figure 2A,2C – it's not clear from figure legend, what exactly does this percentage mean. Weighted by frequency of clonotype or not. Percentage of top N clones, of total repertoire volume?

Data are weighted by the frequency of the clonotypes that were used for analysis and includes all clonotypes with frequencies above the median (see table S1-S5).

We changed legends accordingly.

6. Figure 2b. If n=2, authors should show both mice, not one representative for both animals.

We changed Figure 1b accordingly and rewrote this paragraph on page 5, lines

97-100.

7. Page 6. "TCRbeta seq directly from GC" – what does it mean? Is there a separate step of nucleic acid isolation and library preparation?

Directly from GC means that we did not disrupted the lymph nodes into a cell suspension before RNA isolation and library preparation. To clarify this, we rewrote this sentence on page 6, line 110.

8. Page 6. 1 million of CDR3-containing reads: this number is collected from 1 GC or from the whole lymph node?

The 1 mio TCR sequences were obtained from 4-6 GCs collected by laser microdissection from histological sections to yield sufficient amount of RNA. This information is described on page 5 line 107.

9. Page 6. "This data shows that GC reactions occur simultaneously at two separate lymph nodes independent whether Ag is present or not." Authors should consider moving this phrase forward to the Discussion section or removing it as it is not justified by the data in this paragraph.

To clarify this data, we rewrote this phrase but kept it at the end of this paragraph on page 5 line 113. The histological data clearly show that GC reactions take place in both lymph nodes and in both, the Ag1 and the PBS group.

10. In the literature analysis, authors have missed recent important research in the field of Tfh TCR repertoires, notably https://pubmed.ncbi.nlm.nih.gov/31914381/.

We added this reference in the introduction on page 3, line 59 and in the discussion on page 11, line 262.

11. Figure 1C The phrase "% area of GC of B cell follicles area" is unclear at least for nonprofessionals in imaging: first the area of B cell zone is calculated, and then the percentage of GC?

The cumulative area of the complete B cell zone of one cryosection was determined. Likewise, the cumulative area of all germinal centers of this cryosection was determined. The relative proportion of the germinal center area within the B cell zone area was calculated as quotient. For each sample three individual cryosection were evaluated and the mean proportion was used for evaluation. To clarify this, we included this explanation into the Material and Method section (page 16, line 346) and in the legend of Figure 1.

12. “This data demonstrate that distinct T cell clones contribute to the responses in GC and skins, which is unexpected considering the fact that all T cells were activated with the same Ag.” Why authors imply the unexpected result if the consensus is that Teff and Tfh frequently recognize different set of antigens? Again, this part of discussion may benefit from more recent works on Tfh TCR repertoires.

We apologize for this lack of clarity. We rewrote this paragraph on page 6, line 133

13. Page 7. The similarity of clonotypes is not defined clearly. May be replaced with "sequence overlaps in Tfh TCR repertoires" or "shared identical sequences", or defined through edit distance. It would be useful to see clonotypes tables of all samples to compute different metrics of repertoire pairwise distance. Could authors provide the tables or the raw sequencing data in open access, e.g., through GEO dataset?

Thank you for raising this point. We replaced the phrase “similarity” accordingly. Of course, the data sets will be published in the Sequence Read Archive as done for previous work (bioproject PRJNA586880).

14. Page 9. The usage of sharing metrics in repertoire comparison should be explained in more details. Why a certain sharing index was used, why another edit distance measure is not applicable? Repertoire sharing strongly depends on the size of repertoire, therefore the data cleaning and downsampling needs to be described in details. 18 shared clonotypes and 65 shared clonotypes: are these nucleotide sequences or aminoacid CDR3 sequences? Do they have the same pattern of recombination – is there a match in both TRBV and TRBJ segments used? It's 18 and 65 of 500 clonotypes? Of n clonotypes if sizes of compared repertoires differ?

We apologize for the lack of clarity. We used the term “sharing index” not as a special metrics but just as a simple enumeration of shared Tfh clonotypes between the mice either of the Ag group and or the PBS group as described by Madi et al. 2014, doi:

10.1101/gr.170753.113. First, all the Tfh clonotypes that were detected once per mouse were assessed by combining all Tfh clonotypes of both contralateral lymph nodes and removing all duplicates. These number of unique Tfh clonotypes were combined from all three mice in each group. Thus, 4 wks after immunization, 7449 and 6445 Tfh clonotypes were unique in all three mice of the Ag1 group or PBS group, respectively. From those, 424 (Ag1) or 244 (PBS) Tfh clonotypes were present in 2 mice and 65 versus 18 in 3 mice of the Ag1 group or PBS group. This data demonstrates that Ag-exposure increases the number of the shared Tfh clonotypes between all mice. For a better clarification we rewrote this paragraph on page 8, line 204 and removed the 2 wks and 7 wks groups on Figure 5. For all other samples, the Morisita-Horn-index were used for quantifying shared clonotypes between the groups. This index considers not only the quantity of clonotypes within one sample but also the frequency with which each clonotype exist. We included an explanation on page 6 line128.

15. Figure 5. There are no significant changes on the plot, why the name of the figure strongly claims changes over time? Figure 5B. It's not convincing without statistics that TRBV3 is the most prominent gene family which changes frequency after Ag challenge. Moreover, it is certainly not enough without TCR transfection experiments to claim that this TRBV is specific to the Ag1-derived epitopes. These claims should be removed, or additional TCR cloning and specificity confirmation experiments are required.

Changes in the V/J usage in autoimmune models have been described from other groups before without transfection experiments. In this Figure, we demonstrate that among the Tfh clonotypes that are shared between all three mice especially the percentage of those that bears the TRBV3 gene segment accumulated in the Ag1 group compared to the PBS group (from 16.6% to 33.84%). To make this point clearer, we removed the 2wks and 7wks time points as done for Figure 5A. Due to the fact that only the shared Tfh clonotypes are analyzed statistical analysis is not applicable. Accordingly, we changed our wording and rewrote this paragraph on page 8, line 214.

16. Page 16. "the repertoire of TCR β clonotypes was identified as described above" Should be below instead of above?

Thank you for pointing that out! We changed the wording accordingly.

Reviewer #3 (Significance (Required)):The article "Receptor repertoires of murine follicular T helper cells reveal shared responses in separate lymph nodes" summarizes the results of a beautiful and technically complex study. However, reviewer argues that both the experimental data, the methods, and result presentation need substantial revision and improvement to be considered in a peer-reviewed journal.Reviewer's expertise covers T cell biology, TCRseq data analysis in both human and rodents' immune repertoires datasets, as well as features of different library preparation protocols and sequencing platforms. Unfortunately, TCRseq data analysis in this particular study doesn't stand up to the modern standards and recommendations of AIRR community. Histology and related analysis sections are out of the scope of the reviewer's expertise.

To find out whether the high overlap of Tfh clonotypes in separate pln is biased by the TCR analysis tool MiTCR, which might overestimate TCR repertoire diversities, we reanalyzed the raw data of some key samples with MiXCR and Immunarch according to the reviewer suggestions. Data reveal an almost identical distribution of Tfh clonotypes between both pln regardless of whether MiTCR and MiXCR were used for analysis. Results are included as **Figure 2-figure supplement 1** in the discussion section on page 11 and 12, lines 245-259. In addition, the sharing index, which displays the number of identical (public) Tfh clonotypes between the mice of one group (Figure 5) is now complemented with venn diagrams in supplementary **Figure 2-figure supplement 2.** This information is included in the result section page 10, lines 211-214 and in the Material and Method section page 19 lines 445-446.